# Genome-Wide Analysis of Sugar Transporters Identifies the *gtsA* Gene for Glucose Transportation in *Pseudomonas stutzeri* A1501

**DOI:** 10.3390/microorganisms8040592

**Published:** 2020-04-19

**Authors:** Yaqun Liu, Liguo Shang, Yuhua Zhan, Min Lin, Zhu Liu, Yongliang Yan

**Affiliations:** 1School of Life and Pharmaceutical Sciences, Hainan University, Haikou 570100, China; liuyaqun19900921@126.com; 2Biotechnology Research Institute, Chinese Academy of Agricultural Sciences, Beijing 100081, China; shangliguo12@163.com (L.S.); zhanyuhua@caas.cn (Y.Z.); linmin@caas.cn (M.L.)

**Keywords:** *Pseudomonas stutzeri* A1501, sugar-transport system, glucose, *gtsA* gene, biological nitrogen fixation

## Abstract

*Pseudomonas stutzeri* A1501 possesses an extraordinary number of transporters which confer this rhizosphere bacterium with the sophisticated ability to metabolize various carbon sources. However, sugars are not a preferred carbon source for *P. stutzeri* A1501. The *P. stutzeri* A1501 genome has been sequenced, allowing for the homology-based in silico identification of genes potentially encoding sugar-transport systems by using established microbial sugar transporters as a template sequence. Genomic analysis revealed that there were 10 sugar transporters in *P. stutzeri* A1501, most of which belong to the ATP-binding cassette (ABC) family (5/10); the others belong to the phosphotransferase system (PTS), major intrinsic protein (MIP) family, major facilitator superfamily (MFS) and the sodium solute superfamily (SSS). These systems might serve for the import of glucose, galactose, fructose and other types of sugar. Growth analysis showed that the only effective medium was glucose and its corresponding metabolic system was relatively complete. Notably, the loci of glucose metabolism regulatory systems HexR, GltR/GtrS, and GntR were adjacent to the transporters ABC^MalEFGK^, ABC^GtsABCD^, and ABC^MtlEFGK^, respectively. Only the ABC^GtsABCD^ expression was significantly upregulated under both glucose-sufficient and -limited conditions. The predicted structure and mutant phenotype data of the key protein GtsA provided biochemical evidence that *P. stutzeri* A1501 predominantly utilized the ABC^GtsABCD^ transporter for glucose uptake. We speculate that gene absence and gene diversity in *P. stutzeri* A1501 was caused by sugar-deficient environmental factors and hope that this report can provide guidance for further analysis of similar bacterial lifestyles.

## 1. Introduction

Sugar is a stable and widely distributed carbon source in nature and has traditionally been considered an important source of carbon skeleton material and energy supply in bacteria. Sugar is not the dominant carbon source for *Pseudomonas*, which prefers amino acids or organic acids [1]. The basic sugar biosynthesis and degradation pathways have been established for some time and have more recently become a subject of great interest and enthusiastic study [2,3]. It is perhaps somewhat natural to focus on intracellular activities and forget the preceding step whereby sugar enters the cell via membrane transport. Sugar usually enters the cell through various specific transporters, such as the glucose phosphotransferase transport system (PTS^Glu^) in *Escherichia coli* [4]. Saier summarized about 20 families of secondary carriers, including porins and various transport systems, and constructed the transporter classification database (TCDB) [5]. Due to the limited data in TCDB, we extended our search to include other databases (UniProt and the National Center for Biotechnology Information (NCBI)) and referred to the well-studied model strains of *E. coli* and *Sulfolobus solfataricus* to provide more comprehensive and important information for our research [6].

*Pseudomonas stutzeri* A1501 (China General Microbiological Culture Collection Center Accession No. 0351) was originally isolated from rice rhizosphere soil in South China and was previously named *Alcaligenes faecalis* A15 [7]. It possesses a strong nitrogen fixation ability and can effectively colonize the root surface to promote plant growth [8]. The ability of *P. stutzeri* A1501 to utilize nitrogen has been studied extensively, but little is known about its utilization of carbon sources, especially regarding sugars. We explored the reasons for the growth status of *P. stutzeri* A1501 by considering the sugar-transport system.

The genome of *P. stutzeri* A1501 has been sequenced, enabling the rapid identification and screening of the potential sugar-transport system [9]. In this study, we compiled the sequence data for all sugar transporters present in *P. stutzeri* A1501 using similarity searches and evaluated the potential substrate specificity by analyzing its adjacent metabolism and regulatory genes. Glucose was assessed as the available sugar source by combining growth analysis and metabolic system gene characterization. Hence, the function of the putative glucose transport system was evaluated through the characterization of candidate gene expression under the conditions of sufficient and limited glucose, and their contribution to glucose use was examined using a mutation analysis and consumption experiments.

## 2. Materials and Methods

### 2.1. Bacterial Strains and Growth Conditions

The bacterial strains and plasmids used in this study are shown in Table 1. *P. stutzeri* A1501 was grown in a minimal lactate medium or in a Luria-Bertani (LB) medium at 30 °C as previously described [10]. *E. coli* was grown at 37 °C in the LB medium. Chloramphenicol (Cm), tetracycline (Tc), and kanamycin (Km) were added to the media at concentrations of 20, 10, and 50 μg/mL, respectively.

### 2.2. Computer Analyses and Screening Strategies

All data were based upon sequence comparisons conducted using the freely accessible genome data of *P. stutzeri* A1501. Functional annotations of the protein sequences were carried out by a BLASTP search of the translations versus GenBank’s non-redundant protein database (NR) [9]. To more comprehensively identify the sugar transporter, protein sequences of known sugar-transport systems from the transporter classification database (http://www.tcdb.org/search/index.php) were also used to screen the corresponding sequence of *P. stutzeri* A1501 at the BLAST server of the NCBI (http://ncbi.nlm.nih.gov) with an E-value cutoff of 10^−5^. The identified candidate proteins were then cross-checked against the original list of annotated proteins. To find the possible substrates, the predicted sugar transporters in *P. stutzeri* A1501 were similarity searched by BLASTP alignment in the UniProt database with an E-value cutoff of 10^−5^. Finally, the protein sequence alignments were conducted with the DNAMAN software (Lynnon BioSoft, Vaudreuil, QC, Canada). Candidate sugar-transport protein was speculated to have the same substrate if the protein identity of the *P. stutzeri* A1501 protein was more than a 60% match with the corresponding protein of *Pseudomonas*, or more than a 30% match with other bacteria. Moreover, the coverage should be more than 60% of the protein sequence full-length.

### 2.3. Monitoring Growth

The bacteria were cultured overnight in an LB medium. After centrifugation and washing, they were diluted in the experimental medium to an initial optical density (OD_600)_ of 0.1. The OD_600_ was measured every two hours until the cells reached a steady growth. Three replicates of each strain were performed.

### 2.4. Reverse Transcription PCR (RT-PCR) and Quantitative Real-Time PCR (qPCR)

The total RNA was isolated using the innuPREP RNA Mini Kit (Analytik Jena, Jena, Germany) and reverse transcription was performed using the Prime Script™ RT Reagent Kit with gDNA Eraser (Takara, Dalian, China). The produced cDNA was used to perform qPCR using the ChamQ™ Universal SYBR qPCR Master Mix (Vazyme, Nanjing, China). All primer pairs are shown in Appendix A, and the procedures followed the manufacturer’s instructions. The relative gene expression levels were calculated using the 2^−ΔΔCt^ method and the *16S rRNA* was used as a reference gene.

### 2.5. Phylogenetic Analysis

For the phylogenetic analysis, the amino acid sequences of glucose-binding proteins from different organisms were obtained from the NCBI. Multiple sequence alignments of full-length proteins were performed using ClustalX [14]. The pairwise deletion option was used to circumvent the gaps and missing data. We used the neighbor-joining tree generated by the MEGA (Molecular Evolutionary Genetics Analysis) program [15] with 1000 replicates of bootstrap analysis.

### 2.6. Construction of the gtsA Deletion Mutant and Complementation

To verify the role of *gtsA* gene in *P. stutzeri* A1501, the *gtsA* region was deleted by homologous recombination according to a PCR-based fusion strategy. The upstream and downstream homologous arm fragments of *gtsA* and the chloramphenicol resistance gene *cat* with its own promoter in pKatCAT5 were fused. The fusion PCR product was then cloned into the multiple cloning site of pK18mob*sacB* and the resulting plasmid was named pK18*gtsA*. The obtained recombinant plasmid was transferred into the wild type *P. stutzeri* A1501 by triparental mating, producing a mutant strain. Candidate clones resulting from a double-crossover event were isolated on the LB agar with chloramphenicol (Appendix A). The correct recombination was confirmed by PCR analysis (Appendix A). To complement the *gtsA* gene, a PCR fragment containing *gtsA* was cloned into the plasmid pLAFR3. The resulting vector (pL*gtsA*) was then used to transform the Δ*gtsA* mutant (Appendix A). The correct recombination was confirmed by PCR analysis (Appendix A). The test result was validated by gene sequencing. To compare the phenotype of the complemented strain and the wild type, pLAFR3 was also introduced into *P. stutzeri* A1501.

### 2.7. Measurement of Glucose Uptake

The samples were harvested during the exponential growth of *P. stutzeri* A1501 by filtering the medium. The determination and quantification of glucose were performed using an ion chromatograph (Thermo Scientific™ Dionex™ ICS-5000, Waltham, MA, USA). The gradients of sodium hydroxide, generated by an eluent generator (Thermo Scientific Dionex ICS 5000EG, Waltham, MA, USA), were as follows: 200 mM (0–10 min, equilibration), 200 mM (10–40 min, column cleaning), and 10 mM (40–52 min, sample). Quantitative analyses were performed by measuring the peak area using the external standard method.

## 3. Results

### 3.1. Prediction of Sugar-Transport Systems

The bioinformatics analysis predicted 10 sugar-transport systems in *P. stutzeri* A1501, including the ATP-binding cassette family (ABC), phosphotransferase system (PTS), major facilitator superfamily (MFS), major intrinsic protein family (MIP) and the sodium solute superfamily (SSS) (Appendix A).

The five ABC-type transport systems consisted of one required ATP-binding protein (ABP) and two permeases. With the exception of the transporters encoded by *PST2907*-*2909*, the others also contained one substrate-binding protein (SBP). Although this system had an amino acid sequence identity of 66–83% with the monosaccharide ABC transport system (*PA0136*-*0138*) in *Pseudomonas aeruginosa* [16], the substrate specificity could not be determined due to the absence of SBP. It may be linked to the *E. coli* galactose/methylgalactoside MglABC transport system [17], but the sequence similarity was very low. The proteins encoded by *PST3478*-*3486* and *PST3579*-*3583* share an identity with the maltose transport system *malEFGK* [18] and the ribose transport system *rbsABC* [19] in *E. coli*, respectively. The latter was also similar to the *Haloferax volcanii* DS2 glucose ABC transport system (*HVOB3014*-*3018*). The second potential maltose transporters were encoded by *PST2190*-*2193*, and shared 77–83% similarity with *P. aeruginosa* MtlEFGK [20]. Notably, this system was also similar to the *Burkholderia multivorans* multiple sugar transporter encoded by *BMULJ02554*-*02558* (55–61% identity) and *Aquimixticola soesokkakensis* lactose transporter subunit LacEF encoded by *AQS8620_00608*-*00609* (55–60% identity). The last ABC transport system encoded by *PST2437*-*2440* was similar to the *Pseudomonas putida* glucose transport system GtsABCD_2_ [21], but no corresponding protein was found in *E. coli*.

Only one fructose-specific PTS transport system was discovered. The gene order of *fruB*–*fruK*–*fruA* was the same as in *E. coli*, but the locus of *fruR* was closely linked to this cluster, whereas *E. coli fruR* was expressed independently [4]. The protein encoded by *PST0988* was only 15% identical to *E. coli* FruB, but more than 30% identical to the *Pseudomonas fluorescens* glucose PTS transport protein PtsP (*PFL_4931*) and the *Aquitalea magnusonii* mannose PTS transport protein (*VI06_09940*).

In addition to the aforementioned ABC and PTS sugar-transport systems, the two genes *PST1613* and *PST1972* may be involved in the MFS transport system. The protein encoded by *PST1613* shared 58% identity with the sugar efflux transporter YdeA (also known as SotB) in *E. coli* [22] and 63% with the *Acinetobacter marinus* arabinose export protein. The protein encoded by *PST1972* shared 46% similarity with *Acidovorax* sp. RAC01 MFS/sugar transporter (*A0A1B3PGF1_9BURK*). It also had many functions beyond MFS, such as SSS and glycoside-pentoside-hexuronide (GPH) cation symporter transporter, because it was highly similar not only to melibiose and galactose SSS transporters in *Chitinimonas taiwanensis* and *Janthinobacterium* sp. B9-8, but also to GPH in *Jannaschia helgolandensis.* Another sodium/glucose co-transporter also belongs to the SSS family and was encoded by *PST1574*. Finally, an MIP family protein encoded by *PST1604* had 67% similarity with *E. coli* GlpF [23].

We noticed that the neighborhood of the candidate transport system included sugar-related metabolic genes. The adjacent genes of ABC^GtsABCD^, ABC^MtlEFGK^, ABC^MalEFGK^ and PTS^Fru^ were mainly involved in glucose metabolism, namely *edd-2*/*glk2/hexR/zwf/pgl/eda* (*PST3495*–*PST3550*), *edd-1/glk-1/gltR/gltS* (*PST2441*–*PST2444*), *gntR/gntK* (*PST2198*, *PST2221*) and *gcd* (*PST0991*), respectively. However, the genes around the other systems were not related to sugar metabolism.

Two sugar porins were located upstream of ABC^GtsABCD^ and downstream of ABC^MalEFGK^, which were highly similar to LamB in *E. coli* (>30% identity), so we named them LamB1 and LamB2, respectively [24]. A sugar porin OprB was also present upstream of ABC^GtsABCD^ in *P. aeruginosa* and *P. putida* and was reported to transport glucose into the cell [25]. However, the sequence similarity between LamB1, LamB2, and OprB was very low, and no OprB homologues were found in the *P. stutzeri* A1501 genome.

### 3.2. Growth in Different Sugars

The whole-genome BLAST analysis revealed that *P. stutzeri* A1501 might possess an uptake system for glucose, maltose, mannose, ribose, fructose, xylose, melibiose, lactose, arabinose and galactose. Therefore, growth experiments were performed in a minimal medium (containing all the essential nutrients and trace elements) with sugar as the only carbon source. Surprisingly, with the exception of glucose, *P. stutzeri* A1501 hardly grew on other sugars (Figure 1). We further analyzed the related metabolic pathways of the sugars in cytosol to explore the reasons.

Firstly, glucose was converted to glucose-6-phosphate and fructose-6-phosphate by glucokinase Glk and glucose-6-phosphate isomerase Pgi. Fructose was phosphorylated into fructose-1,6-diphosphate by fructokinase MtlZ and 1-phosphofructokinase FruK [26]. These three products eventually entered the Entner–Doudoroff (ED) pathway, pentose phosphate (PPP) pathway and the tricarboxylic acid (TCA) cycle. The pattern diagram and regulatory genes were published elsewhere and will not be repeated here [9].

The PPP pathway involved the conversion between xylulose-5-phosphate, ribose-5-phosphate, and ribulose-5-phosphate. However, xylose, arabinose and ribose could not be metabolized through the PPP pathway because xylose isomerase XlyA [27], arabinose isomerase AraA [28] and ribokinase Rbsk [29] were not found in *P. stutzeri* A1501. Similarly, even though mannose-6-phosphate could be converted from fructose-6-phosphate by mannose-6-phosphate isomerase AlgA, there was no hexokinase to phosphorylate mannose [30]. Maltodextrin phosphorylase (MalP) and amylomaltase (MalQ) were essential enzymes for maltose and maltodextrin metabolism [18], galactose dehydrogenase (GalDH) and 2-dehydro-3-deoxy-6-phosphate galactate aldolase (DgoA) were equally important for galactose [31] and only MalQ and GalDH were present in this organism. We did not find any proteins related to lactose and melibiose metabolism. In short, only glucose and fructose metabolic systems were relatively complete metabolic systems, whereas others were incomplete or non-existent (Figure 2).

### 3.3. Effect of Glucose on the Expression of Sugar Transporters

We discussed the influence of *P. stutzeri* A1501 when the substrate was glucose for the following four reasons: one system can interact with multiple substrates; the candidate transporters neighborhood genes are related to glucose metabolism; this bacterium can only grow in glucose and has a more complete glucose metabolic system.

The expression level of related genes may vary with glucose concentration to maintain glucose homeostasis. Therefore, we analyzed the expressions of the genes of these candidate sugar-transport systems under glucose-sufficient (25 mmol/L) and -limited (3 mmol/L) conditions to confirm their hypothetical role. RNA was isolated rapidly after shaking the culture for 1 h because the growth of *P. stutzeri* A1501 was limited in the absence of a carbon source or low concentration of glucose and long-term culture would affect RNA quality.

Table 2 shows that both concentrations of glucose could significantly induce the mRNA expression of *PST2437*, *PST2438*, *PST2439*, and *PST2440*, especially at 25 mmol/L. The *PST3484*, *PST1972*, *PST3079*, *PST3685,* and *PST1221* mRNA levels were upregulated only with 25 mmol/L glucose, but not with 3 mmol/L. The other sugar transporter genes were not affected regardless of the concentration.

### 3.4. Functional Analysis of Glucose-Binding Protein GtsA

A striking result was the strong induction of the ABC^GtsABCD^ transport system by different concentrations of glucose, so we focused on the *gtsA* gene which encodes the periplasmic binding protein, a key component of this system. The genomic sequence analysis showed that *gtsA* (*PST2440*) was located upstream from *gtsBCD/lamB1* (*PST2436*–*PST2439*, encoding permease, ABP, and porin) and transcribed in the same orientation, suggesting that they may be co-transcribed. This possibility was determined using the indicated primer pairs to amplify cDNA and genomic DNA (gDNA). The results showed that the PCR products of *gtsB*–*gtsC* were obtained using both cDNA and gDNA as templates. However, *gtsA*–*gtsB* produced a PCR product only with gDNA (Figure 3A). This result indicated that *gtsA* was independently transcribed in *P. stutzeri* A1501.

Th phylogenetic analysis revealed that the product of *gtsA* was highly conserved in *Pseudomonas* species and was most closely related to *Ochrobactrum anthropi* glucose-binding protein (oaGBP) [32]. Compared with *E. coli* GBP, GtsA was more closely related to the corresponding protein from extreme environmental microorganisms such as *Thermus thermophilus* ttGBP [33], *Saccharolobus solfataricus* GlcS [6], and *Thermotoga maritima* GBP [34] (Figure 3B). Although X-ray crystallographic analysis was not performed, the tertiary structure of GtsA was inferred by homology-modeling based on the crystal structure of *P. putida* CSV86 glucose-binding protein (ppGBP; Protein Data Bank code 5DVI) [32] (Figure 3C; Appendix A).

To study the biological function of *gtsA* gene, a knockout mutant was constructed by replacing the *gtsA* gene with a chloramphenicol resistance cassette. The growth and consumption curve of the strains in the medium containing glucose as the sole carbon source showed that the mutant strain Δ*gtsA* could hardly grow (Figure 4A) and the glucose content in the medium did not change (Figure 4B), which meant that the strain could not utilize glucose. The results also showed that the complemented strain Δ*gtsA* (pL*gtsA*) could restore the above phenotypes while the complementary plasmid pLAFR3 had no effect on the strain.

## 4. Discussion

All free-living bacteria in nature can rapidly and flexibly adapt their behavior to the changing environment, thereby effectively utilizing the available carbon source. Sugar, the most common nutrient, plays a vital role in microbial metabolism. For example, it is the main carbon source for *E. coli*, which is rich in sugar-transport systems and metabolic pathways. Although *Pseudomonas* are ubiquitous bacteria, unlike *E. coli*, sugar is not their preferred carbon source [35]. *P. aeruginosa* prefers succinic acid rather than glucose and fructose [36]. The poor sugar use may be due to catabolite repression control [37] or related genes loss, such as the well-known deficiency of the glycolysis key enzyme phosphofructokinase [38].

In this study, a total of 10 limited and incomplete potential sugar-transport systems in *P. stutzeri* A1501 were identified by bioinformatics and genome analysis. This was consistent with a previous report that only fructose enters into *Pseudomonas* through the PTS system, whereas other sugars are independent of PTS [35]. The sugar metabolism system of *P. stutzeri* A1501 was also deficient. There was only one relatively complete glucose metabolism system, and other sugar degrading enzymes that participated in the central carbon metabolism were lacking. L-rhamnose isomerase RHI (*Q75WH8_PSEST*) can metabolize rhamnose, mannose, ribose, and many other sugars in *Pseudomonas* sp. LL172 [39,40]; it is strange that its homologues are found in *Rhizobium* and not *Pseudomonas*. Fewer genes for sugar metabolism were present in *P. stutzeri* A1501 compared with other *Pseudomonas* species, such as ribokinase RbsK (*PFLU_4156*) which was present in *P. fluorescens* but not in *P. stutzeri* A1501. This phenomenon may be influenced by the living environment; *P. fluorescens* was isolated from the phyllosphere [41], whereas *P. stutzeri* A1501 was isolated from the rhizosphere, which has a lower sugar content [42,43]. Previous studies suggested that endophytic *Pseudomonas* could utilize the sugars in xylem fluid in contrast to the rhizosphere *Pseudomonas* [44]. Based on their distinct material conditions, microorganisms will selectively discard some “useless” genes in their long-term adaptive evolution process and prioritize genes that are conducive to competitive survival [45]. Studies generally have reported that environmentally induced gene loss is an adaptive evolutionary ability of bacteria to avoid invalid gene replication, transcription and translation, to save energy and to improve space efficiency [46].

Of all the assayed sugars, *P. stutzeri* A1501 only grew on glucose, which may be due to it having a more complete system for glucose transport and metabolism. Different concentrations of glucose had varying degrees of influence on the transport system; what attracted our attention is that ABC^GtsABCD^ could be significantly induced regardless of the concentration, indicating that it participated in glucose transport. This system has been deeply studied in *P. putida* and *P. aeruginosa* [21]. A *gtsA* deletion mutant was constructed to further clarify the function, and *P. stutzeri* A1501 almost completely lost the ability to utilize glucose after inferring with the function of the ABC^GtsABCD^ transport system. Glucose transport and chemotaxis were also defective in the absence of glucose-binding proteins in *P. aeruginosa* [47].

Glucose was internalized into *E. coli* by the ABC^MglABC^, the PTS^Glu^ system, and GalP [48]; its homologues GalP (*PST4041*) and ABC^MglAC^ (*PST2907*–*PST2908*, <30% identity) were found in *P. stutzeri* A1501, but the substrate-binding protein MglB and the PTS^Glu^ transport system were absent. The same situation exists in *Archaea* isolated from extreme environments, such as *Sulfolobus solfataricus* [49], *Thermophilic archaea* [50] and *Thermotoga maritima* [51]. They lack the PTS^Glu^ system and rely on the ABC transport system [52], which is more similar to ABC^GtsABCD^ than ABC^MglABC^. We speculate that this genetic diversity is guided by different survival circumstances, and the microorganism shows the optimum phenotype to cater to its surroundings, which is the basis for a higher-level evolution of species [53]. For *E. coli*, glucose was sufficient, and its transport system was more comprehensive and powerful, whereas the ABC transport system was more suitable for sugar-deficient bacterium such as *Pseudomonas* and *Archaea*. The ABC transporter is considered essential for thermophiles to survive in harsh habitats because it has a stronger binding capacity than other systems and can bind to very low concentrations of substrates [54].

At present, the sugar-transport systems of *Pseudomonas* have not been systematically analyzed. We hope that the summary provided in this article can guide the further development of related research, which needs to be improved and supplemented continuously, even though the genome of *P. stutzeri* A1501 has been comprehensively analyzed. *P. stutzeri* A1501 lacks many sugar-related genes, but retains some genes related to glucose metabolism whose functional roles deserve attention.

## Figures and Tables

**Figure 1 microorganisms-08-00592-f001:**
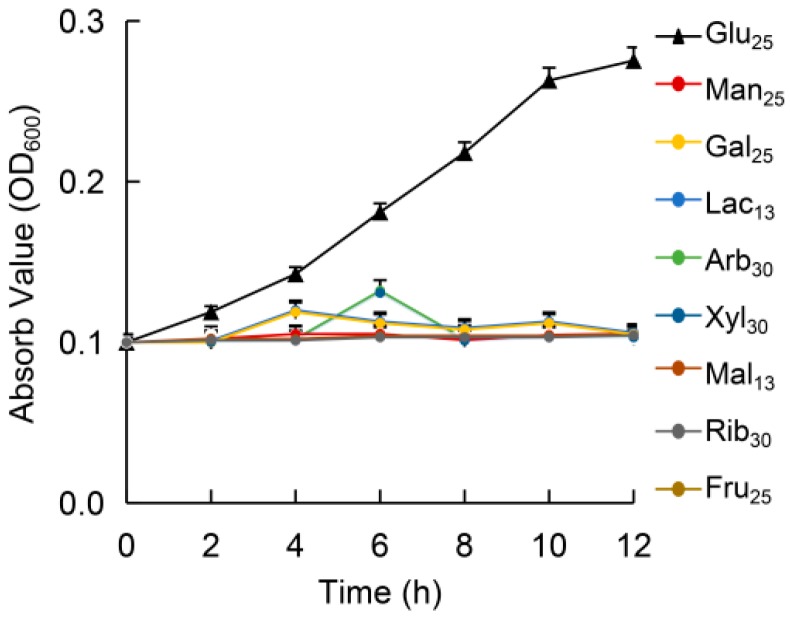
Growth of *Pseudomonas stutzeri* A1501 in a minimal medium containing different sugars with the same number of carbon atoms (glucose, 25 mmol/L; mannose, 25 mmol/L; galactose, 25 mmol/L; lactose, 13 mmol/L; arabinose, 30 mmol/L; xylose, 30 mmol/L; maltose, 13 mmol/L; ribose, 30 mmol/L; and fructose, 25 mmol/L). The values are the means of three independent experiments. Error bars represent the standard deviation (SD) of the three biological replicates from a single experiment. For some data points, the SDs were smaller than the symbol size, so the error bars are indiscernible.

**Figure 2 microorganisms-08-00592-f002:**
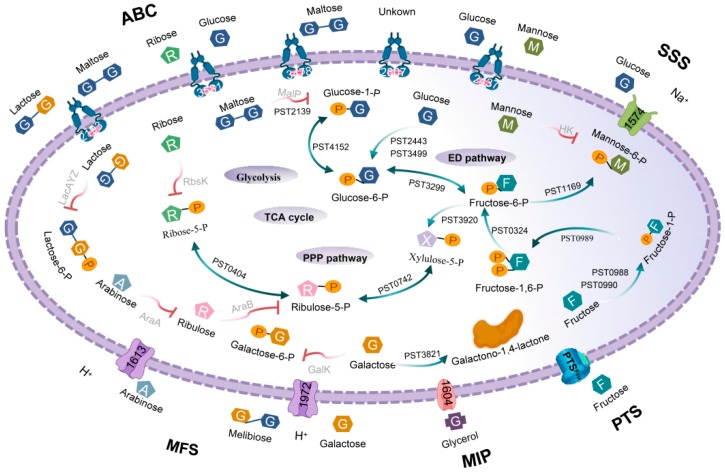
The substrate and metabolic pathways of the inferred sugar-transport system in *P. stutzeri* A1501. Detailed gene annotation is provided in Appendix A.

**Figure 3 microorganisms-08-00592-f003:**
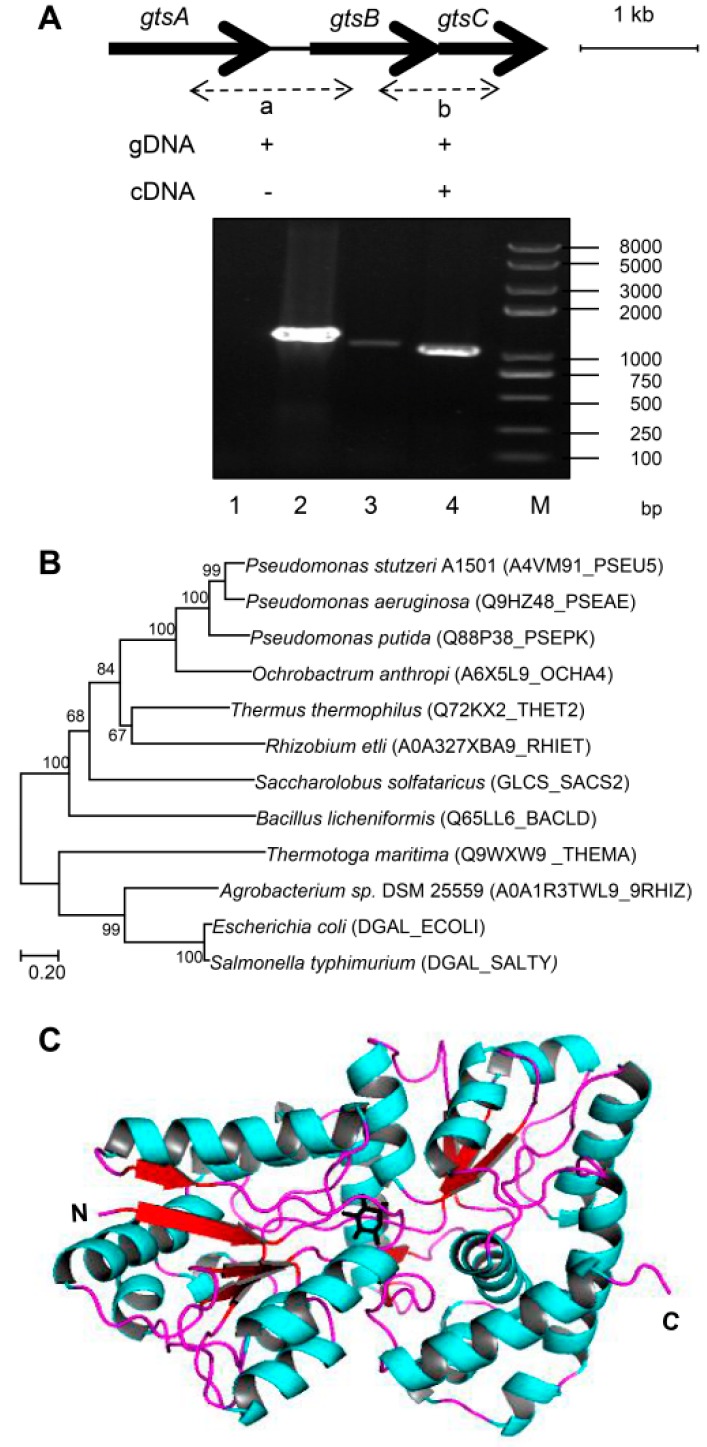
Gene organization of *gtsABC* operon, phylogenetic analysis and proposed structure of GtsA. (**A**) Independent transcription of the *gtsA* locus genes determined with RT-PCR. The *gtsA–gtsB* (a, lanes 1 and 2) and *gtsB–gtsC* (b, lanes 3 and 4) junctions were amplified using genomic DNA (gDNA, lanes 2 and 4) and cDNA (lanes 1 and 3) as the template. Lane M, 2 kb plus DNA ladder and the sizes of the molecular markers are indicated at the side in bp. (**B**) Unrooted neighbor-joining phylogenetic tree of the *P. stutzeri* A1501 GtsA was constructed after the multiple alignment of data by ClustalX [14]. Bootstrap values based on 1000 replications are listed as percentages at branching points. (**C**) Homology model of the GtsA involved in the regulation of glucose transport pathways. The N- and C-terminal domains are shown, and glucose is represented by a black stick. The figure was generated using Swiss-Model (https://swissmodel.expasy.org/) and PyMOL (http://www.pymol.org).

**Figure 4 microorganisms-08-00592-f004:**
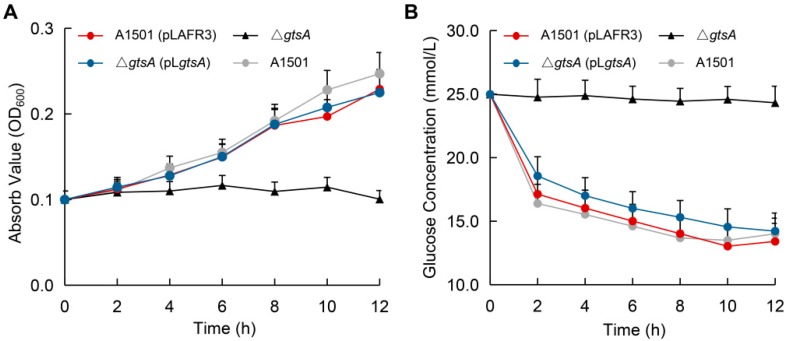
(**A**) Growth and (**B**) substrate consumption curves of the Δ*gtsA* mutant strain in the glucose-sufficient medium (25 mmol/L).

**Table 1 microorganisms-08-00592-t001:** Strains and plasmids used in this study.

Strains or Plasmids	Relevant Characteristics	Source or Reference
**Strains**		
*P. stutzeri* A1501	Wild type, Chinese Culture Collection: CGMCC (China General Microbiological Culture Collection Center) 0351	[10]
Δ*gtsA*	*gtsA* deleted mutant strain, Cm^r^	This study
Δ*gtsA* (pL*gtsA*)	Δ*gtsA* complemented with pL*gtsA*, Km^r^, Tc^r^	This study
A1501 (pL*gtsA*)	A1501 complemented with pL*gtsA*, Km^r^	This study
**Plasmids**		
pLAFR3	Broad host range cloning vector, Tc^r^	[11]
pK18mob*sacB*	Suicide plasmid for gene knockout, Km^r^	[12]
pRK2013	Used as mobilizing plasmid in triparental crosses, Km^r^	[13]
pK18*gtsA*	Deleted *gtsA* fragment cloned into pK18mob*sacB*, Km^r^, Cm^r^	This study
pL*gtsA*	pLAFR3 derivative carried a fragment encoding the *gtsA* gene, used to complement, Tc^r^	This study

**Table 2 microorganisms-08-00592-t002:** Sugar-transport system of *P. stutzeri* A1501 and the effect of glucose on related genes.

Family/Substrate	Locus Tag(s)	Fold Change ^a^		Homologue/References *
		Glucose_25_	Glucose_3_	
**ATP-binding Cassette Family**				
Maltose/Mannitol/Lactose	*PST2190*-*2193*	NSS ^b^	NSS ^b^	*mtlEFGK, P. aeruginosa* [20]
Glucose/Mannose	*PST2437*-*2440*	+727.96	+24.21	*gtsABCD*, *P. putida* [21]
		+181.73	+25.23	
		+241.87	+81.08	
		+222.30	+42.64	
Unknown	*PST2907*-*2909*	NSS ^b^	NSS ^b^	
Maltose/Maltodextrin/Maltooligosaccharide	*PST3478*	+2.33	NSS ^b^	*malEFGK, E. coli* [18]
	*PST3484*-*3486*	NSS ^b^		
Ribose/Glucose	*PST3579*-*3583*	NSS ^b^	NSS ^b^	*rbsABC*, *E. coli* [19] *tsgABCD13*, *H. volcanii* DS2
**Phosphotransferase System**				
Fructose	*PST0987*-*0990*	NSS ^b^	NSS ^b^	*fruAB*, *E. coli* [4]
**Major Facilitator Superfamily**				
Sugar/Arabinose	*PST1613*	NSS ^b^	NSS ^b^	*ydeA*, *E. coli* [22]
Melibiose/Galactose	*PST1972*	+2.81	NSS ^b^	
**Major Intrinsic Protein Family**				
Glycerol	*PST1604*	NSS ^b^	NSS ^b^	*glpF*, *E. coli* [23]
**Sodium Solute Superfamily**				
Glucose	*PST1574*	NSS ^b^	NSS ^b^	

* The selected strains and references are representative, and homologues of these transporters have been found in other microorganisms that might interact with other substrates (Appendix A). ^a^ Relative mRNA expression levels of the sugar-transport system genes under different glucose concentrations (glucose_25_: 25 mmol/L; glucose_3_: 3 mmol/L). Data in the ABC transport system are sorted by gene order. ^b^ Not statistically significant.

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
