# Peer review of "Genome-Wide Analysis of Sugar Transporters Identifies the gtsA Gene for Glucose Transportation in Pseudomonas stutzeri A1501"

_microorganisms, 2020, doi:10.3390/microorganisms8040592_

Round 1
Reviewer 1 Report
Dear authors,
Bioinformatics analyses musy be improved, you should protein sequences of predicted 10 sugar transport systems in P. stutzeri A1501, power of nucleotide sequences are weak than protein sequences !!
Thus, bioinformatics analyses must be redo with protein sequences of predicted 10 sugar transport systems in P. stutzeri A1501
"The sequence homology greater than 30% is presumed to have the same substrate. " what mean? nucleotide or protein sequences? If nocleotide sequences, 30% is very low threshold!
Why did not perform alignments for 10 sugar transport systems? Only was done for GtsA with ppGBP in Supp. Fig. 2
Author Response
Dear reviewer,
Thank you for your suggestions and comments. We have carefully considered all the comments and accordingly revised the manuscript. Moreover, our manuscript has been revised by a professional English editing service (MDPI).
We hope that the revised version is suitable for publication.
Sincerely yours
Yongliang Yan, PhD
Professor
Biotechnology Research Institute, Chinese Academy of Agricultural Sciences
Beijing, China
E-mail: [email protected]
Zhu Liu, PhD
Professor
School of Life and Pharmaceutical Sciences, Hainan University
Haikou 570100, China
E-mail: [email protected]
1 Bioinformatics analyses must be improved, you should protein sequences of predicted 10 sugar transport systems in P. stutzeri A1501, power of nucleotide sequences are weak than protein sequences !! Thus, bioinformatics analyses must be redo with protein sequences of predicted 10 sugar transport systems in P. stutzeri A1501
A: Thank you for bringing this issue to our attention. In this work, protein sequences were used for the bioinformatics analyses. We apologize for not making this clear in the former version and have revised the manuscript accordingly. All data are based upon sequence comparisons conducted using the freely accessible genome data of P. stutzeri A1501. Protein sequences were sampled and subjected to similarity searches (BLASTP) using the BLAST server of the National Center for Biotechnology Information (NCBI) with an E-value cutoff of 10-5. The identified candidate proteins were then cross-checked against the original list of annotated proteins. To find the possible substrates, predicted sugar transporters in P. stutzeri A1501 were similarly searches by BLASTP alignment to the UniProt database with an E-value cutoff of 10-5.
Now, computer analyses and screening strategies have been described in greater details (pages 2-3, lines 70-84 in the revised version), and protein sequence similarity analysis has also been identified in the results (pages 4-5, lines 126-170 in the revised version). Figure S3 is also marked as protein sequence alignment of sugar transport system components of P. stutzeri A1501 and other microorganisms (Supplementary Materials, page4, line 31 in the revised version).
2 "The sequence homology greater than 30% is presumed to have the same substrate. " what mean? nucleotide or protein sequences? If nocleotide sequences, 30% is very low threshold!
A: Sequence homology here refers to “protein sequence homology”, which was revised in the methods and results section. It was read as: Protein sequence alignments were conducted with DNAMAN software (Lynnon BioSoft, Vaudreuil, QC, Canada). Candidate sugar transport proteins were speculated to have the same substrate if protein identity of the P. stutzeri A1501 protein was more than 60% to the corresponding protein of Pseudomonas, or more than 30% to other bacteria. And the coverage should be more than 60% of the protein sequence full-length. (page 3, lines 79-84 in the revised version).
3 Why did not perform alignments for 10 sugar transport systems? Only was done for GtsA with ppGBP in Supp. Fig. 2
A: Growth analysis showed that only glucose can be utilized by P. stutzeri A1501 in minimal medium, and the glucose metabolism system of the bacterium was relatively complete. Thus, we focused on the effects of glucose on these ten sugar transport systems and found that only ABCGtsABCD expression was significantly upregulated under both glucose-sufficient and -limited conditions. GtsA is a key substrate-binding protein in this transport system, and it is 64% identity to glucose-binding protein (ppGBP, Protein Data Bank code 5DVI) in P. putida CSV86, so we only speculated the tertiary structure of GtsA by homology modeling.

Reviewer 2 Report
This study by Liu et al. investigates sugar transporters in Pseudomonas stuzeri A1501, A rhizobacterial strain. They performed BLAST search and various other analyses (e.g., phylogeny, expression, glucose update and bacterial growth) and conclude that this bacterial strain mostly uses ABC transporters for glucose uptake, possibly related to its presumably glucose-deficient environment.
In general, I think this is useful attempt to understand the biology of this bacterial strain. The manuscript for the most part is clearly written and the generated data are sound in general. I have the following comments for the authors to consider in their revision.
1. The manuscript frequently lacks technical details. For instance, it is unclear what evalue cutoff was used in their BLAST search. No any details were given for phylogenetic analyses (e.g., information of phylogenetic methods used in analyses, rate heterogeneity, bootstrap analyses etc).
2. There are too many errors in the manuscript. Please see the following for some random examples:
Line 100, “similar search” should be “similarity search”.
Line 103, “The sequence homology greater than 30%”. This is incorrect. There is no such thing as 30% homology. Two sequences are either homologous or non-homologous. I assume that the authors meant 30% sequence identity or similarity
Lines 132-141. Please see above for comments on homology. All these errors should be corrected.
Lines 224-225, Figure 3B. It does not make a lot of sense to classify this small number of sequences into subfamilies. Moreover, the tree is unrooted, how do you determine the root and classify the subfamilies?
Author Response
Dear reviewer,
Thank you for your suggestions and comments. We have carefully considered all the comments and accordingly revised the manuscript. Moreover, our manuscript has been revised by a professional English editing service (MDPI). The all response to your comments is in the attachment, please check it.
We hope that the revised version is suitable for publication.
Sincerely yours
Yongliang Yan, PhD
Professor
Biotechnology Research Institute, Chinese Academy of Agricultural Sciences
Beijing, China
E-mail: [email protected]
Zhu Liu, PhD
Professor
School of Life and Pharmaceutical Sciences, Hainan University
Haikou 570100, China
- mail: [email protected]
This study by Liu et al. investigates sugar transporters in Pseudomonas stuzeri A1501, A rhizobacterial strain. They performed BLAST search and various other analyses (e.g., phylogeny, expression, glucose update and bacterial growth) and conclude that this bacterial strain mostly uses ABC transporters for glucose uptake, possibly related to its presumably glucose-deficient environment.
In general, I think this is useful attempt to understand the biology of this bacterial strain. The manuscript for the most part is clearly written and the generated data are sound in general. I have the following comments for the authors to consider in their revision.
Specific critiques
1 The manuscript frequently lacks technical details. For instance, it is unclear what evalue cutoff was used in their BLAST search. No any details were given for phylogenetic analyses (e.g., information of phylogenetic methods used in analyses, rate heterogeneity, bootstrap analyses etc).
A: Thank you for the reminder. We have completed the corresponding technical methods. It was read as: 2.2. Computer Analyses and Screening Strategies
All data are based upon sequence comparisons conducted using the freely accessible genome data of P. stutzeri A1501. Functional annotations of protein sequences were carried out by BLASTP search of the translations versus GenBank’s non-redundant protein database (NR) [9]. To more comprehensively identify the sugar transporter, protein sequences of known sugar transport systems from the transporter classification database (http://www.tcdb.org/search/index.php) were also used to screen the corresponding sequence of P. stutzeri A1501 at the BLAST server of the National Center for Biotechnology Information (NCBI) with an E-value cutoff of 10-5. The identified candidate proteins were then cross-checked against the original list of annotated proteins. To find the possible substrates, predicted sugar transporters in P. stutzeri A1501 were similarity searches by BLASTP alignment to the UniProt database with an E-value cutoff of 10-5. Finally, protein sequence alignments were conducted with DNAMAN software (Lynnon BioSoft, Vaudreuil, QC, Canada). Candidate sugar transport protein was speculated to have the same substrate if protein identity of the P. stutzeri A1501 protein was more than 60% to the corresponding protein of Pseudomonas, or more than 30% to other bacteria. And the coverage should be more than 60% of the protein sequence full-length (pages 2-3, lines 69-84 in the revised version).
2.5. Phylogenetic Analysis
For phylogenetic analysis, the amino acid sequences of GtsA proteins from different organisms were obtained from the NCBI (http://ncbi.nlm.nih.gov). Multiple sequence alignments of full-length proteins were performed using ClustalX [14]. The pairwise deletion option was used to circumvent the gaps and missing data. We used the neighbor-joining tree generated by the MEGA program [15] with 1000 replicates of bootstrap analysis (page 3, lines 97-102 in the revised version).
2 There are too many errors in the manuscript.
A:The manuscript has been professionally edited by MDPI English editors
3 Please see the following for some random examples: Line 100, “similar search” should be “similarity search”.
A: Thanks for the reviewer; the “similar search” was replaced by “similarity searches” (page 3, line 78 in the revised version).
4 Line 103, “The sequence homology greater than 30%”. This is incorrect. There is no such thing as 30% homology. Two sequences are either homologous or non-homologous. I assume that the authors meant 30% sequence identity or similarity
A: Thank you for bringing this issue to our attention. We have replaced the term “sequence homology” with “protein identity” (page 3, lines 81 in the revised version) and the term “sequence similarity” (page 4, lines 135 in the revised version).
5 Lines 132-141. Please see above for comments on homology. All these errors should be corrected.
A: We have corrected “are homologous to” to “share identity with” (page 4, line 136 in the revised version), “is 58% homologous to” to “shares 58% identity with” (page 4, line 152 in the revised version) and “is highly homologous” to “is highly similar” (page 4, line 155 in the revised version).
6 Lines 224-225, Figure 3B. It does not make a lot of sense to classify this small number of sequences into subfamilies. Moreover, the tree is unrooted, how do you determine the root and classify the subfamilies?
A: In order to depict the genetic and evolutionary distances of GtsA homologous proteins in different species, an unrooted neighbor-joining phylogenetic tree of GtsA homologous proteins was constructed after multiple alignment of data by ClustalX. In order to well-documented the detailed information of this work, a description of “Phylogenetic Analysis” was added in the methods section (page 3, lines 97-102 in the revised version).
We also modified the result section, it was read as: Phylogenetic analysis revealed that GtsA is highly conserved in Pseudomonas species and is most closely related to Ochrobactrum anthropi oaGBP [32]. Compared with E. coli GBP, GtsA is more closely related to the corresponding protein from extreme environmental microorganisms such as Thermus thermophilus ttGBP [33], Saccharolobus solfataricus GlcS [6], and Thermotoga maritima GBP [34] (Figure 3B) (page 7, lines 239-243 in the revised version).

Reviewer 3 Report
The authors have identified and further characterized the genes (10 sugar transporters) potentially encoding sugar transport systems based on the genome in Pseudomonas stutzeri A1501 using well established methodology such as gene knockout and complementation. I believe that the authors have provided sufficient background, presented the results and concluded appropriately based on available data. However, I have some suggestions on its presentation of results that I would like to recommend to the authors for consideration. Specifically,
Line 35, “…in recent years.” need references.
Line 42, delete “etc.”
Line 45, “…A15” needs references.
Line 47-49, rewrite, this sentence is confusing.
Line 67, the table needs to be numbered., also in this table, in the first column, the word “plasmids” should not be bold; alternatively, the word “strain” should be bold.
Line 68, I would like to suggest that the authors make a figure to show the constructions of gene knockout and complementation mutants, and how the successful constructions of these mutations are confirmed.
Line 107, delete the first word “In”
Line 126, a space is needed in “E.coli” and many other cases, e.g., in Table 1, throughout the entire manuscript.
Figure 3A, lanes of the gel need to be labelled, and the markers on the right lane missi ng unit (bp?)
Figure 3B: The phylogenetic tree is presented but the construction of the phylogenetic tree was not explained in Materials and Methods.
Line 240, remove “,”
Lines 273-274, rewrite, this sentence is confusing.
In addition, there are many editorial errors that need to corrected throughout the entire manuscript. For example, there are many cases that the latin names and gene names need to be italicized.
Author Response
Dear reviewer,
Thank you for your suggestions and comments. We have carefully considered all the comments and accordingly revised the manuscript. Moreover, our manuscript has been revised by a professional English editing service (MDPI). The all response to your comments is in the attachment, please check it.
We hope that the revised version is suitable for publication.
Sincerely yours
Yongliang Yan, PhD
Professor
Biotechnology Research Institute, Chinese Academy of Agricultural Sciences
Beijing, China
E-mail: [email protected]
Zhu Liu, PhD
Professor
School of Life and Pharmaceutical Sciences, Hainan University
Haikou 570100, China
E-mail: [email protected]
The authors have identified and further characterized the genes (10 sugar transporters) potentially encoding sugar transport systems based on the genome in Pseudomonas stutzeri A1501 using well established methodology such as gene knockout and complementation. I believe that the authors have provided sufficient background, presented the results and concluded appropriately based on available data. However, I have some suggestions on its presentation of results that I would like to recommend to the authors for consideration.
Specific critiques
1 Line 35, “…in recent years.” need references.
A: We would like to thank the Reviewer for this suggestion. The followed two references were added to the main text. [1](Antonovsky, N.; Gleizer, S.; Noor, E.; Zohar, Y.; Herz, E.; Barenholz, U.; Zelcbuch, L.; Amram, S.; Wides, A.; Tepper, N.; et al. Sugar synthesis from CO2 in Escherichia coli. Cell. 2016, 166, 115–125)and [2] (Thattai, M.; Shraiman, B.I. Metabolic switching in the sugar phosphotransferase system of Escherichia coli. Biophys. J. 2003, 85, 744–754.) (page 1, line 37 and page 10, lines 335-338 in the revised version).
2 Line 42, delete “etc.”
A: We have modified this sentence as suggested (page 2, line 44 in the revised version).
3 Line 45, “…A15” needs references.
A: Here we cited the followed reference. Verimeiren, H.; Willems, A.; Schoofs, G.; Mot, R. De; Keijers, V.; Hai, W.L.; Vanderleyden, J. The rice inoculant strain Alcaligenes faecalis A15 is a nitrogen-fixing Pseudomonas stutzeri. Syst. Appl. Microbiol. 1999, 22, 215–224. (page 2, line 48 and page 10, lines 345-347 in the revised version).
4 Line 47-49, rewrite, this sentence is confusing.
A: The sentence has been revised into “The ability of P. stutzeri A1501 to utilize nitrogen has been studied extensively, but little is known about its utilization of carbon sources, especially regarding sugars” (page 2, lines 49-51 in the revised version).
5 Line 67, the table needs to be numbered., also in this table, in the first column, the word “plasmids” should not be bold; alternatively, the word “strain” should be bold.
A: Thank you for pointing this issue out. This table has been numbered as “Table 1” (page 2, lines 64 and 68 in the revised version), and the subsequent tables have also been renumbered accordingly (Table 2, page 6, lines 218 and 223 in the revised version). At the same time, the word “plasmids” and “strains” in Table 1 were bolded (page 2, line 68 in the revised version).
6 Line 68, I would like to suggest that the authors make a figure to show the constructions of gene knockout and complementation mutants, and how the successful constructions of these mutations are confirmed.
A: To address this comment, the section of the gtsA deletion mutant construction (page 3, lines 103-116 in the revised version) was substantially revised, and new Figure S1 and Figure S2 (Supplementary Materials, page2, lines 14-28 in the revised version) on the construction and validation of the gtsA mutant and ΔgtsA (pLgtsA) were added in the Materials and Methods. The subsequent supplementary figure have also been renumbered accordingly (It was marked in the manuscript and Supplementary Materials).
2.6. Construction of the gtsA Deletion Mutant and Complementation
To verify the role of GtsA in P. stutzeri A1501, the gtsA region was deleted by homologous recombination according to a PCR-based fusion strategy. The upstream and downstream homologous arm fragments of gtsA and the chloramphenicol resistance gene cat with its own promoter in pKatCAT5 were fused. The fusion PCR product was then cloned into the multiple cloning site of pK18mobsacB, and the resulting plasmid was named pK18gtsA. The obtained recombinant plasmid was transferred into the wild type P. stutzeri A1501 by triparental mating, producing a mutant strain. Candidate clones resulting from a double-crossover event were isolated on LB agar with chloramphenicol (Figure S1A). Correct recombination was confirmed by PCR analysis (Figure S1B). To complement the gtsA gene, a PCR fragment containing gtsA was cloned into the plasmid pLAFR3. The resulting vector (pLgtsA) was then used to transform the ΔgtsA mutant (Figure S2A). Correct recombination was confirmed by PCR analysis (Figure S2B). The test result was validated by gene sequencing. To compare the phenotype of the complemented strain and the wild type, pLAFR3 was also introduced into P. stutzeri A1501.
The Figure S1 is in the attachment. Please check it.
Figure S1. Construction and validation of gtsA deletion mutant. (A) Schematic representation of the gtsA deletion mutant generated by replacing the gtsA region with the chloramphenicol resistance gene cat (Cmr). The primer pairs testF1–testR1 and testF2–testR2 were used to analyze the gtsA deletion as indicated by arrows and the corresponding sequences are shown in Table S1. (B) Validation of gtsA deletion mutant by colony PCR. The testF1–testR1 (lanes 2 and 4) and testF2–testR2 (lanes 1 and 3) junctions were amplified using wild type strain (lanes 3 and 4) and gtsA deletion mutant (lanes 1 and 2) as the template. Lane M, 15 kb plus DNA ladder, and the sizes of the molecular markers are indicated at the side in bp.
The Figure S2 is in the attachment. Please check it.
Figure S2. Construction and validation of ΔgtsA (pLgtsA). (A) Schematic representation of the ΔgtsA (pLgtsA) generated by introducing pLgtsA into the gtsA deletion mutant. The primer pair testF3–testR3 was used to analyze the ΔgtsA (pLgtsA) and the corresponding sequences are shown in Table S1. (B) Validation of ΔgtsA (pLgtsA) by colony PCR (lane 1). Lane M, 15 kb plus DNA ladder, and the sizes of the molecular markers are indicated at the side in bp.
7 Line 107, delete the first word “In”
A: Done, We have modified “In silico analysis” to “Bioinformatics analysis” (page 4, line 127 in the revised version).
8 Line 126, a space is needed in “E.coli” and many other cases, e.g., in Table 1, throughout the entire manuscript.
A: Thank you for pointing this mistake out. We have replaced the word "E.coli" with “E. coli” (page 4, lines 137, 144, 146 and 147; page 6, line 223 and page 9, line 304 in the revised version).
9 Figure 3A, lanes of the gel need to be labelled, and the markers on the right lane missing unit (bp?)
A: We have added the annotations to the modified Figure 3A (page 8, line 253 in the revised version).
The Figure 3A is in the attachment. Please check it.
Figure 3. (A) Independent transcription of the gtsA locus genes determined with RT-PCR. The gtsA–gtsB (a, lanes 1 and 2) and gtsB–gtsC (b, lanes 3 and 4) junctions were amplified using genomic DNA (gDNA, lanes 2 and 4) and cDNA (lanes 1 and 3) as the template. Lane M, 2 kb plus DNA ladder, the sizes of the molecular markers are indicated at the side in bp.
10 Figure 3B: The phylogenetic tree is presented but the construction of the phylogenetic tree was not explained in Materials and Methods.
A: Thank you for pointing out these issues. We have added a description of phylogenetic analysis (page 3, lines 97-102 in the revised version) as follows:
2.5. Phylogenetic Analysis
For phylogenetic analysis, the amino acid sequences of GtsA proteins from different organisms were obtained from the NCBI (http://ncbi.nlm.nih.gov). Multiple sequence alignments of full-length proteins were performed using ClustalX [14]. The pairwise deletion option was used to circumvent the gaps and missing data. We used the neighbor-joining tree generated by the MEGA program [15] with 1000 replicates of bootstrap analysis.
11 Line 240, remove “,”
A: Done, We have modified “GtsA, was” to “GtsA was” (page 8, line 259 in the revised version).
12 Lines 273-274, rewrite, this sentence is confusing.
A: The sentence has been revised into “Studies generally report that environmentally induced gene loss is an adaptive evolutionary ability of bacteria to avoid invalid gene replication, transcription, and translation; save energy; and improve space efficiency” (page 9, lines 292-294 in the revised version).
13 In addition, there are many editorial errors that need to corrected throughout the entire manuscript. For example, there are many cases that the latin names and gene names need to be italicized.
A: We have carefully checked the main text and italicized all the Latin names and gene names as suggested (It was marked in the revised version). The manuscript has also been professionally edited by MDPI English editors.

Round 2
Reviewer 1 Report
Dear authors;
All requirements were made carerully.
Best wishes..
Reviewer 3 Report
I appreciate very much the efforts that the authors have devoted to making significant improvement to their manuscripts. I am satisfied with the revision and have no more comment on it.